# USP21 prevents the generation of T-helper-1-like Treg cells

Yangyang Li[1,*], Yue Lu[2,*], Shuaiwei Wang[1], Zhijun Han[3], Fuxiang Zhu[1], Yingmeng Ni[4], Rui Liang[1], Yan Zhang[5], Qibin Leng[6], Gang Wei[3], Guochao Shi[4], Ruihong Zhu[7], Dan Li[1], Haikun Wang[8], Song Guo Zheng[9,10], Hongxi Xu[2], Andy Tsun[1,11] & Bin Li[1,12,13]

FOXP3[+] Regulatory T (Treg) cells play a key role in the maintenance of immune homeostasis and tolerance. Disruption of *Foxp3* expression results in the generation of instable Treg cells and acquisition of effector T-cell-like function. Here we report that the E3 deubiquitinase USP21 prevents the depletion of FOXP3 at the protein level and restricts the generation of T-helper-1-like Treg cells. Mice depleted of *Usp21* specifically in Treg cells display immune disorders characterized by spontaneous T-cell activation and excessive T-helper type 1 (Th1) skewing of Treg cells into Th1-like Treg cells. USP21 stabilizes FOXP3 protein by mediating its deubiquitination and maintains the expression of Treg signature genes. Our results demonstrate how USP21 prevents FOXP3 protein depletion and controls Treg lineage stability *in vivo*.

[1] Key Laboratory of Molecular Virology and Immunology, CAS Center for Excellence in Molecular Cell Science, Unit of Molecular Immunology, Institut Pasteur of Shanghai, Shanghai Institutes for Biological Sciences, Chinese Academy of Sciences, Shanghai 200025, China. [2] School of Pharmacy, Shanghai University of Traditional Chinese Medicine, Shanghai 201203, China. [3] Chinese Academy of Sciences-Max Planck Society (MPG) Partner Institute for Computational Biology, Shanghai Institutes for Biological Sciences, Chinese Academy of Sciences, 320 Yueyang Road, Shanghai 200031, China. [4] Department of Pulmonary Medicine, Rui Jin Hospital, School of Medicine, Shanghai Jiao Tong University, Shanghai 200025, China. [5] Key Laboratory of Molecular Virology and Immunology, Unit of Hematopoietic Stem Cell and Transgenic Animal Model, Institut Pasteur of Shanghai, Shanghai Institutes for Biological Sciences, Chinese Academy of Sciences, Shanghai 200025, China. [6] Key Laboratory of Molecular Virology and Immunology, Unit of Immune Regulation, Institut Pasteur of Shanghai, Shanghai Institutes for Biological Sciences, Chinese Academy of Sciences, Shanghai 200025, China. [7] Flow Cytometry Core Facility, Institut Pasteur of Shanghai, Shanghai Institutes for Biological Sciences, Chinese Academy of Sciences, Shanghai 200025, China. [8] Key Laboratory of Molecular Virology and Immunology, Unit of the Regulation of Immune Cell Differentiation, Institut Pasteur of Shanghai, Shanghai Institutes for Biological Sciences, Chinese Academy of Sciences, Shanghai 200025, China. [9] Clinical Immunology Center, Third Affiliated Hospital at Sun Yat-Sen University, Guangzhou 510630, China. [10] Division of Rheumatology, Department of Medicine, Penn State University Hershey College of Medicine, Hershey, Pennsylvania 17033, USA. [11] Innovent Biologics (Suzhou) Co., Ltd, 168 Dongping Street, Suzhou Industrial Park, Suzhou, Jiangsu Province 215123, China. [12] Shanghai Institute of Immunology, Shanghai JiaoTong University School of Medicine, Shanghai 200025, China. [13] Department of Immunology and Microbiology, Shanghai JiaoTong University School of Medicine, Shanghai 200025, China. * These authors contributed equally to this work. Correspondence and requests for materials should be addressed to B.L. (email: binli@sibs.ac.cn).

FOXP3$^+$ Regulatory T (Treg) cells have immune suppressive capacity and are crucial for the maintenance of immune homeostasis and control of dominant immune tolerance[1–3]. Therefore, it is of paramount importance to understand the mechanism underlying lineage stability of Treg cells in vivo. Debate over the stability of FOXP3$^+$ Treg cells has arisen from disparate conclusions of previous studies[4–8]. In these studies, instable Treg cells acquire a T-effector-cell-like phenotype in response to inflammatory or lymphopenic cues, and also found to have instable FOXP3 expression[5,9–11]. Instable Treg cells produce higher amounts of inflammatory cytokines, which positively correlates with autoimmunity[5,10,12,13].

The immunosuppressive phenotype of Treg cells is largely determined by gene expression patterns driven by FOXP3 (refs 2,3,14). Compromised epigenetic programing at the Foxp3 gene locus abrogates its gene transcription and facilitates the generation of exFOXP3 T cells[5,15–18]. These exFOXP3 T cells may produce inflammatory cytokines that lead to the rapid onset of autoimmune diseases[5,10]. In addition to the transcriptional control of the Foxp3 gene, the stability of FOXP3 expression is also determined at the post-translational level. For example, Treg cells respond to stress signals elicited by proinflammatory cytokines and lipopolysaccharides by degrading FOXP3 protein to then acquire a T-effector-cell-like phenotype[19–21]. Thus, the direct tracing of FOXP3 protein and its stability in vivo would contribute to the better understanding of instable Treg cells and their physiological role in health and disease.

Usp21 conventional knockout mice develop splenomegaly and spontaneous T-cell activation[22,23], suggesting a potential role of USP21 in maintaining immune tolerance. We previously identified how the E3 deubiquitinase USP21 is highly induced in human CD4$^+$CD25$^{hi}$CD127$^{lo}$ Treg cells from asthma patients[24], but the in vivo function of USP21 remained unclear.

To illustrate the function of USP21 in vivo, we generate mice with conditional depletion of Usp21 in Treg cells to investigate the role of USP21 in controlling Treg-cell stability. We find that mice lacking USP21 in Treg cells suffer from immune disorders characterized by spontaneous T-cell activation and excessive T-helper type 1 (Th1) skewing. Moreover, Treg-specific deletion of Usp21 leads to significant induction of Th1-like Treg cells. USP21 stabilizes FOXP3 protein by mediating its deubiquitination and maintains the expression of Treg signature genes. Taken together, our results show that USP21 prevents FOXP3 protein depletion and controls Treg lineage stability in vivo.

## Results

**Depletion of Usp21 in Treg cells perturbs immune tolerance.** To illustrate the function of USP21 in controlling Treg-cell fate in vivo, we developed a mouse model where Usp21 is depleted only in Treg cells (Usp21$^{fl/fl}$Foxp3$^{Cre}$ mice) by crossing Usp21$^{fl/fl}$ mice with mice bearing YFP-fused Cre recombinase under control of the Foxp3 gene locus (Fig. 1a). We first analysed thymic development of CD4$^+$ and CD8$^+$ T cells, and no significant difference was observed between Foxp3$^{Cre}$ and Usp21$^{fl/fl}$Foxp3$^{Cre}$ mice (Supplementary Fig. 1a,b). Meanwhile, we did not find significant changes in the absolute numbers of CD4$^+$CD8$^-$ (CD4-SP), CD4$^-$CD8$^+$ (CD8-SP), CD4$^-$CD8$^-$ (DN) and CD4$^+$CD8$^+$ (DP) thymocytes in Usp21$^{fl/fl}$Foxp3$^{Cre}$ mice (Supplementary Fig. 1c). Usp21$^{fl/fl}$Foxp3$^{Cre}$ mice developed lymphadenopathy and splenomegaly at the age of 6 to 8 months (Fig. 1b,c), suggesting aberrant immune activation. We next tested whether Treg-specific deletion of Usp21 perturbed T-cell activation and homeostasis. We observed increased frequency of CD62L$^{lo}$CD44$^{hi}$ effector memory T cells in Usp21$^{fl/fl}$Foxp3$^{Cre}$ mice (Fig. 1d,e). We also observed excessive Th1 responses in Usp21$^{fl/fl}$Foxp3$^{Cre}$ mice, where splenic CD4$^+$YFP$^-$ effector

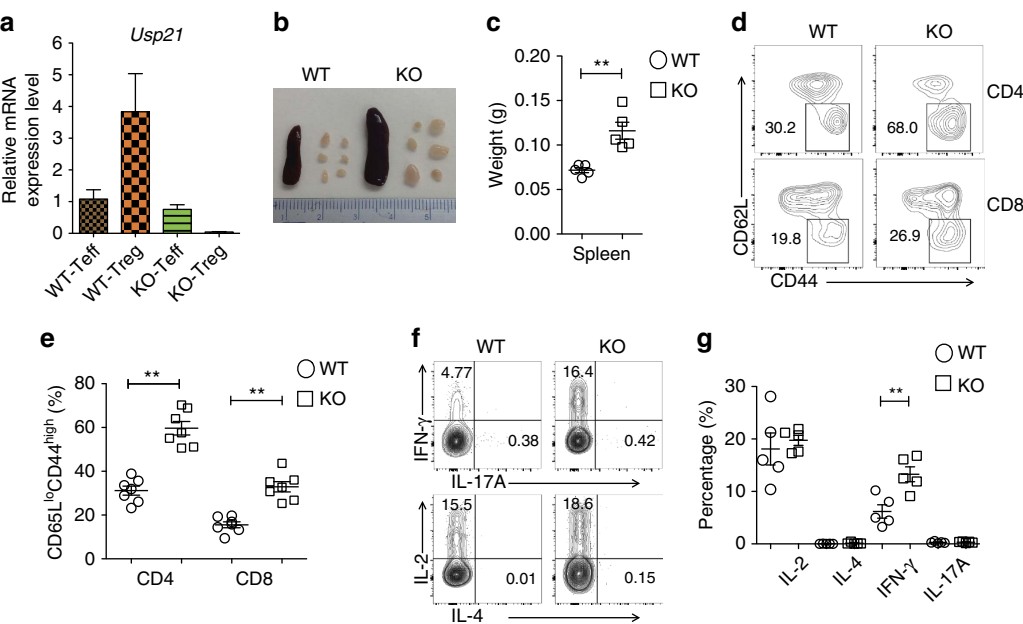

**Figure 1 | Usp21$^{fl/fl}$Foxp3$^{Cre}$ mice develop spontaneous lymphoproliferative disease. (a)** CD4$^+$CD25$^-$YFP$^-$ effector T (Teff) cells and CD4$^+$CD25$^{hi}$YFP$^+$ Treg cells were sorted from Foxp3$^{Cre}$ (WT) and Usp21$^{fl/fl}$Foxp3$^{Cre}$ (KO) mice. mRNA expression of Usp21 in each population was assessed by qRT–PCR. (n = 3 for each group). **(b)** Image of spleens and peripheral lymph nodes (pLNs) from 8-month-old WT and KO mice. **(c)** Quantitative analysis of the weight of spleens isolated from WT (n = 5) and KO (n = 5) mice. **(d)** Representative figure shown the expression of CD62L and CD44 in splenic CD4$^+$YFP$^-$ and CD8$^+$ T cells from WT (n = 7) and KO (n = 7) littermates. **(e)** Percentage of splenic CD62L$^{lo}$CD44$^{hi}$ effector memory T cells as in **d**. **(f)** Representative figure shown the expression of IFN-γ, IL-17, IL-2 and IL-4 by splenic CD4$^+$YFP$^-$ effector T (Teff) cells from WT (n = 5) and KO (n = 5) mice. **(g)** Percentage of IFN-γ$^+$, IL-17$^+$, IL-2$^+$ and IL-4$^+$ splenic T cells as in **f**. Small horizontal lines indicate the mean ( ± s.d.). All data represent means ± s.d. **P ≤ 0.01, as determined by Student's t-test.

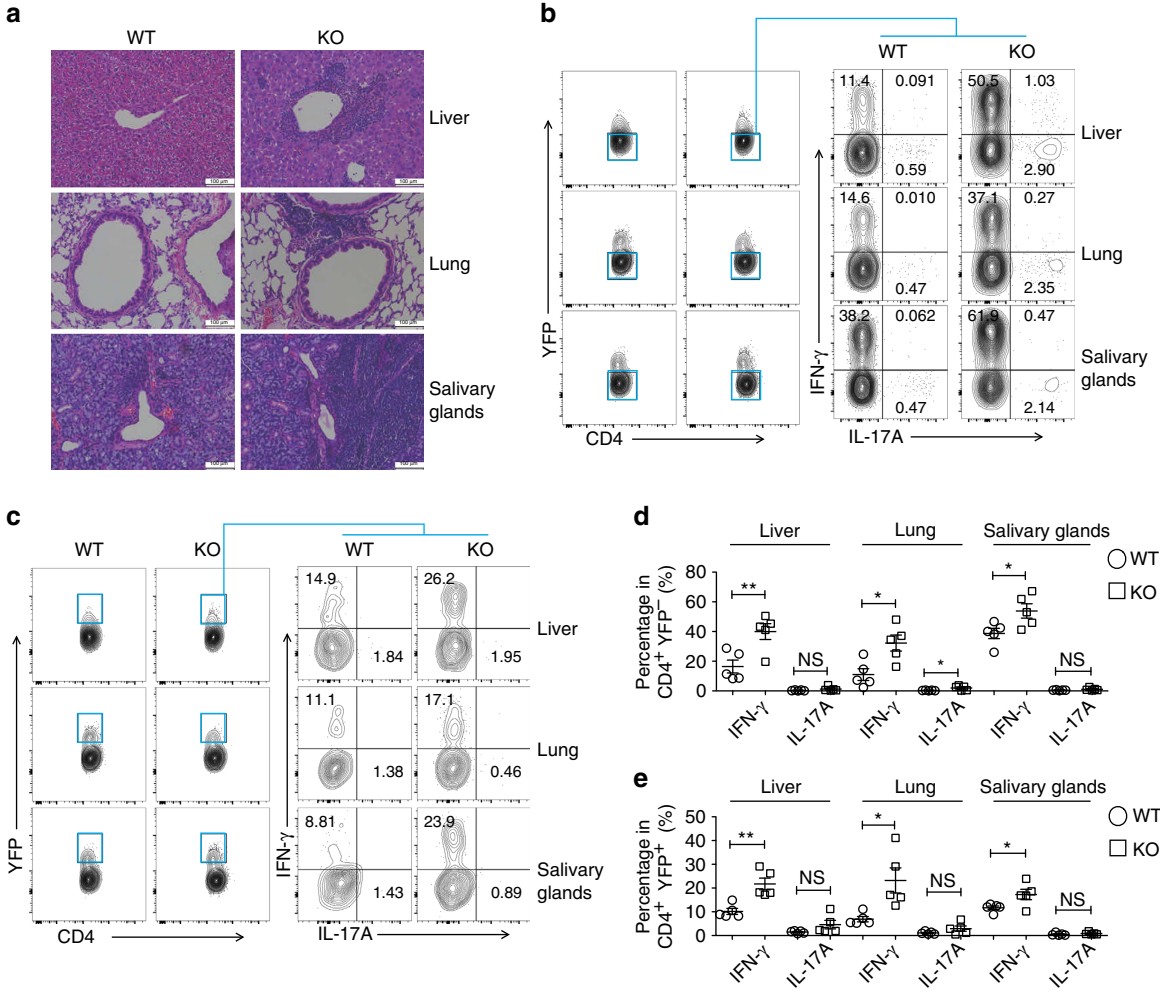

**Figure 2 | USP21-ΔTreg cells acquire a Th1-like phenotype.** (**a**) Histological analysis of lymphocytic infiltration in the liver, lung and salivary glands of indicated mice by H&E staining. Scale bar, 100 μm. (**b**) Representative figure shown the expression of IFN-γ and IL-17A among CD4⁺YFP⁻ effector T (Teff) cells in the liver, lung and salivary glands of WT ($n = 5$) and KO ($n = 5$) mice. (**c**) Representative figure shown the expression of IFN-γ and IL-17A among CD4⁺YFP⁺ Treg cells in the liver, lung and salivary glands of WT ($n = 5$) and KO ($n = 5$) mice. (**d**) Percentage of IFN-γ⁺ and IL-17A⁺ Teff cells as shown in **b**. (**e**) Percentage of IFN-γ⁺ and IL-17A⁺ Treg cells as shown in **c**. All data represent means ± s.d. *$P ≤ 0.05$, **$P ≤ 0.01$, as determined by Student's *t*-test. NS, not significant.

T cells (Teff) produced higher levels of interferon (IFN)-γ on *in vitro* stimulation (Fig. 1f,g). Therefore, USP21-deficient Treg cells failed to maintain immune tolerance and the related inflammation *in vivo*.

**USP21-ΔTreg cells acquire a Th1-like phenotype.** Further histological analysis revealed increased lymphocytic infiltration into peripheral organs of *Usp21*^fl/fl^*Foxp3*^Cre^ mice, including the liver, lung and salivary glands (Fig. 2a). We next analysed the expression of inflammatory cytokines produced by T effector and Treg cells at inflamed tissue loci. We observed the increased production of IFN-γ by CD4⁺YFP⁻ effector T cells in the liver, lung and salivary glands of *Usp21*^fl/fl^*Foxp3*^Cre^ mice (Fig. 2b,d), revealing aberrant immune activation and severe selective Th1-type inflammation. Although the expression of IL-17A slightly increased in the lung of *Usp21*^fl/fl^*Foxp3*^Cre^ mice, its expression was much lower than IFN-γ (Fig. 2b,d). We found that inflamed tissue-derived CD4⁺YFP⁺ USP21-ΔTreg cells produced high amounts of IFN-γ and displayed a Th1-cell-like phenotype (Fig. 2c,e).

Naive CD4⁺ T cells were further sorted from wild type (WT) and *Usp21*^fl/fl^*Foxp3*^Cre^ mice and differentiated into Th1, Th17 and iTreg cells under polarizing conditions. We observed an increased production of IFN-γ in iTreg cells from *Usp21*^fl/fl^ *Foxp3*^Cre^ mice (Supplementary Fig. 2a), confirming that USP21-ΔTreg cells displayed a Th1-like phenotype. However, the production of both IFN-γ and IL-17A was comparable in polarized Th1 and Th17 cells from WT and *Usp21*^fl/fl^*Foxp3*^Cre^ mice (Supplementary Fig. 2a). Therefore, Th1 skewing in *Usp21*^fl/fl^*Foxp3*^Cre^ mice is potentially due to aberrant function of Th1-like USP21-ΔTreg cells. In summary, we observed a population of Th1-like USP21-ΔTreg cells, which correlated with Th1-type inflammation in the liver, lung and salivary glands of *Usp21*^fl/fl^*Foxp3*^Cre^ mice.

**USP21 prevents the loss of FOXP3 protein in Treg cells.** To investigate the stability of USP21-ΔTreg cells, we next analysed the expression of FOXP3 using flow cytometry. We observed the downregulation of FOXP3 protein in USP21-ΔTreg cells from lymphoid as well as non-lymphoid organs (Fig. 3a), while the *Foxp3* gene was still actively transcribed (Supplementary Fig. 3a). This suggested that the loss of USP21 affected the post-translational modification-mediated degradation of FOXP3 protein in these USP21-ΔTreg cells. Further testing indicated that

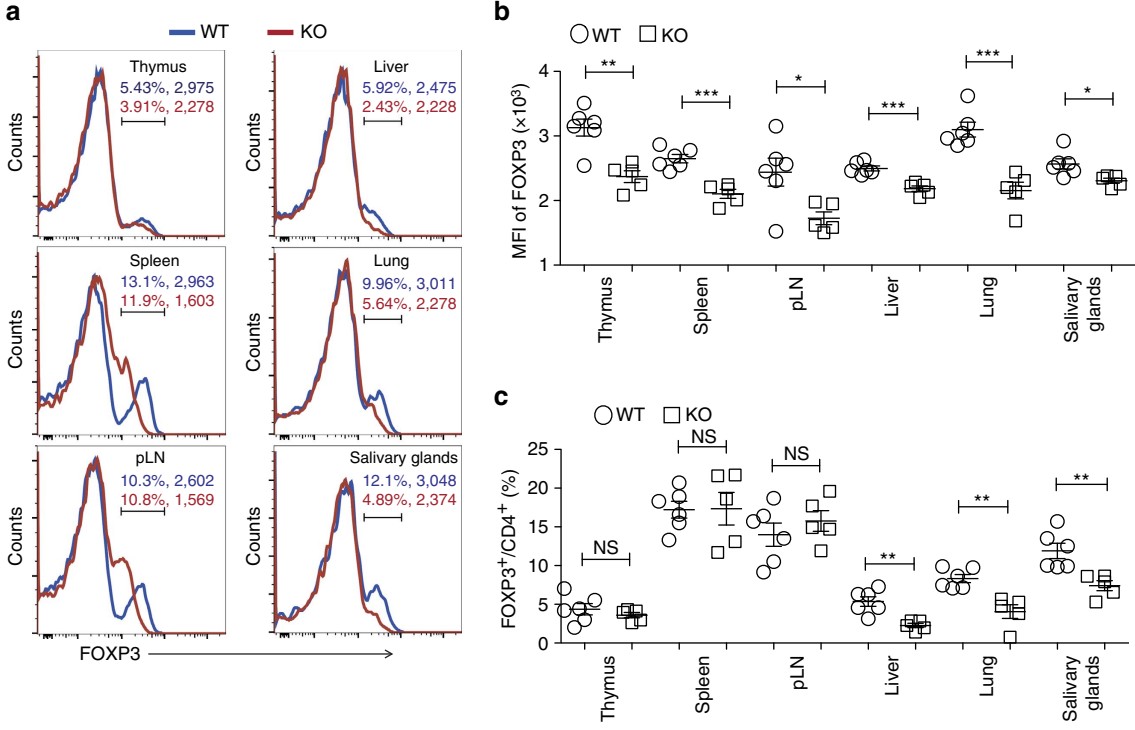

**Figure 3 | Instability of FOXP3 protein in USP21-ΔTreg cells. (a)** Representative figure shown the expression of FOXP3 protein in CD4[+] T cells from the thymus, spleen, pLNs, liver, lung and salivary glands of WT ($n = 6$) and KO ($n = 5$) mice. FOXP3 expression in CD4[+] T cells was shown as histograms and FOXP3[+] cells were further gated. The percentage of CD4[+]FOXP3[+] T cells and mean fluorescent intensity (MFI) of FOXP3 from WT and KO mice were shown in blue (WT) and red (KO). **(b)** MFI of FOXP3 among FOXP3[+] Treg cells indicated in **a**. **(c)** Percentage of FOXP3[+] Treg cells as indicated in **a**. All data represent means ± s.d. *$P \leq 0.05$, **$P \leq 0.01$, ***$P \leq 0.001$, as determined by Student's t-test. NS, not significant.

USP21 is required to stabilize FOXP3 protein, since the mean fluorescence intensity of FOXP3 staining was downregulated in USP21-ΔTreg cells (Fig. 3b). More importantly, the percentages of CD4[+]YFP[+] USP21-ΔTreg cells remained unaffected (Supplementary Fig. 3b–d), reflecting a normal distribution of Treg cells in the lymphoid as well as non-lymphoid organs of *Usp21*[fl/fl] *Foxp3*[Cre] mice. Decrease in the percentages of FOXP3[+] Treg cells was more pronounced in non-lymphoid organs, due to increased loss of FOXP3 protein expression (Fig. 3c). We next checked the expression of FOXP3 in CD4[+]YFP[+] Treg cells and confirmed the instability of FOXP3 protein in CD4[+]YFP[+] USP21-ΔTreg cells (Fig. 4a,b). To further test the *in vivo* stability of USP21-ΔTreg cells, we transferred WT Treg or USP21-ΔTreg cells into Rag1[−/−] mice. There was a significant loss of FOXP3 in *in vivo* transferred USP21-ΔTreg cells (Fig. 4c–e). Taken together, these results indicated that USP21 might control Treg lineage stability by preventing the loss of FOXP3 protein.

**Depletion of *Usp21* perturbs Treg signature gene expression**. We performed RNA sequencing and compared gene expression profiles of Treg cells from *Usp21*[fl/fl]*Foxp3*[Cre] and *Foxp3*[Cre] mice. We first compared differentially expressed genes (DEGs) of Treg and Teff signature genes[25]. We found that USP21-ΔTreg cells displayed impaired transcription of Treg signature genes and simultaneously acquired the expression of genes controlling Teff cell fate (Fig. 5a,b), supporting that USP21-ΔTreg cells became Teff-like. Moreover, in WT and USP21-ΔTreg cells, 35.4% (87 upregulated genes and 64 downregulated genes) of 426 DEGs were FOXP3 direct targets (Fig. 5c), where total FOXP3 targeting genes were taken from published CHIP-seq data[14]. The expression of representative genes was further validated by qRT–PCR (Supplementary Fig. 4a). In summary, these data

suggest that USP21 regulates the function of Treg cells majorly through FOXP3.

**USP21-ΔTreg cells display impaired suppressive activity**. We next examined the suppressive activity of USP21-ΔTreg cells using an *in vitro* suppression assay. We found that USP21-ΔTreg cells had significantly impaired suppressive capacity towards Teff cell proliferation (Fig. 5d,e). Knockdown of *Usp21* in WT Treg cells also impaired their suppressive activity (Supplementary Fig. 4b–d), confirming that USP21 is required for Treg-cell function. These results collectively suggest that USP21 maintains the expression of Treg signature genes and controls the suppressive function of Treg cells.

We further challenged *Usp21*[fl/fl]*Foxp3*[Cre] mice with MOG peptide in an experimental allergic encephalomyelitis (EAE) disease model. We found that *Usp21*[fl/fl]*Foxp3*[Cre] mice developed more severe EAE symptoms and failed to restrict late stages of disease development (Fig. 6a; Table 1). Disease severity was accompanied by increased production of IFN-γ and IL-17A in CD4[+]YFP[−] effector T cells from *Usp21*[fl/fl]*Foxp3*[Cre] mice (Fig. 6b,c). We then transferred CD45.2[+] WT Treg or USP21-ΔTreg cells into EAE-bearing CD45.1 mice, and CD45.2[+] USP21-ΔTreg cells became instable through FOXP3 loss (Fig. 6d,e). Therefore, USP21-ΔTreg cells fail to limit EAE disease development at later phases of disease progression, possibly due to the loss of FOXP3 expression and suppressive function.

**USP21 interacts with FOXP3 in Treg cells**. Our data suggested that a post-translational modification-mediated mechanism controls the degradation of FOXP3 in USP21-ΔTreg cells (Fig. 3a; Supplementary Fig. 3a). To test whether USP21 is a direct E3

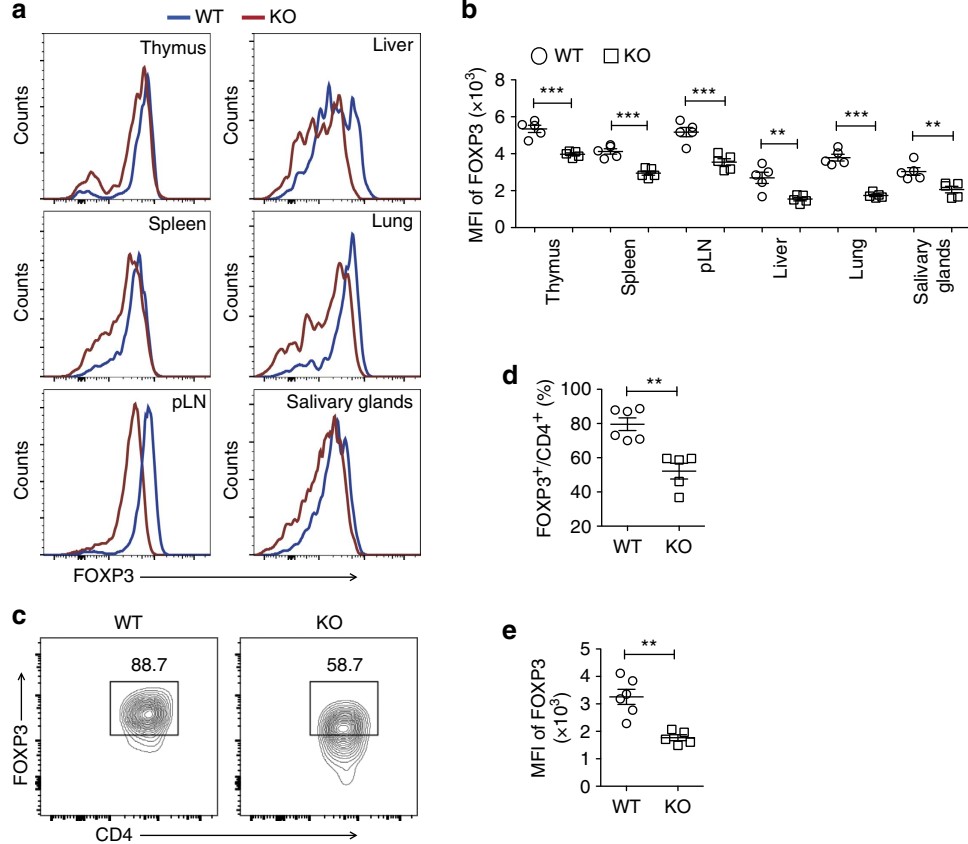

**Figure 4 | Instability of FOXP3 protein in CD4$^+$YFP$^+$ USP21-ΔTreg cells.** (**a**) Representative figure shown the expression of FOXP3 protein in CD4$^+$YFP$^+$ Treg cells from the thymus, spleen, pLNs, liver, lung and salivary glands of WT ($n=5$), and KO ($n=5$) mice. FOXP3 expression in CD4$^+$YFP$^+$ T cells was shown as histograms. (**b**) Mean fluorescent intensity (MFI) of FOXP3 among CD4$^+$YFP$^+$ Treg cells indicated in **a**. (**c**) WT Treg or USP21-ΔTreg cells were transferred into Rag1$^{-/-}$ mice. The expression of FOXP3 protein was further analysed in transferred Treg cells from the spleen of Rag1$^{-/-}$ mice 7 days later. (**d**) Percentage of FOXP3$^+$ Treg cells as indicated in **c**. WT group ($n=6$) and KO group ($n=5$), data represent means ± s.d. (**e**) Mean fluorescent intensity (MFI) of FOXP3 among transferred Treg cells as indicated in **c**. WT group ($n=6$) and KO group ($n=5$), data represent means ± s.d. **$P \le 0.01$, ***$P \le 0.001$, as determined by Student's $t$-test. NS, not significant.

deubiquitinase of FOXP3 that prevents its degradation, we first carried out binding studies to determine whether FOXP3 could directly bind to USP21. Reciprocal immunoprecipitation of FLAG-FOXP3 and Myc-USP21 revealed an interaction between USP21 and FOXP3 (Fig. 7a), and we also detected the endogenous interaction between USP21 and FOXP3 in human Treg cells (Fig. 7b). We next expressed and purified His-USP21, MBP and MBP-FOXP3 from *Escherichia coli*, and carried out an *in vitro* MBP pulldown assay that confirmed the direct interaction between FOXP3 and USP21 (Fig. 7c). Through the generation of systematic deletion mutants (Supplementary Fig. 5a) and co-IP experiments, we found that the zinc-finger subdomain of FOXP3 was essential for its interaction with USP21; deletion of the zinc-finger regions (N2, C1 and C2) disrupted their interaction (Supplementary Fig. 5b). These data further suggested a direct role of USP21 in Treg cells through interaction with FOXP3.

**USP21 deubiquitinates FOXP3 in Treg cells.** We further found the loss of FOXP3 protein in USP21-ΔTreg cells could be prevented by the addition of proteasome inhibitor MG132 (Fig. 7d), suggesting that the ubiquitin-proteasome pathway might be involved in this process. We next studied the half-life of FOXP3 using the protein synthesis inhibitor cycloheximide. And ectopically expressed USP21 significantly extended the half-life

of FOXP3 protein (Fig. 7e). Therefore, USP21 prevents the proteasomal degradation of FOXP3 in Treg cells.

Using a his-tagged ubiquitin pulldown assay, we found that ectopically expressed USP21, but not the enzymatically inactive mutant C221A, reduced levels of ubiquitinated FOXP3 (Fig. 7f). We also observed a much higher level of endogenous ubiquitinated FOXP3 in USP21-ΔTreg cells (Fig. 7g), correlating with the loss of FOXP3 protein *in vivo*. Therefore, these results support that USP21 prevents FOXP3 depletion in Treg cells through deubiquitination and protection from ubiquitination-mediated protein degradation.

To identify the specific lysine residues of FOXP3 that could be deubiquitinated by USP21, we screened each individual lysine-only mutants of FOXP3 (where all lysines in FOXP3 were mutated to arginines besides one residue) and identified seven lysine residues (K206, K216, K227, K252, K277, K332 and K393), which were potential USP21 targets. We then tested mutants where only seven lysine residues were retained or mutated into arginines (termed 7K and 7R mutants, respectively, Supplementary Fig. 5c). His-ubiquitin pulldown assays further confirmed that USP21 could deubiquitinate FOXP3 at these seven lysine residues and that the 7R construct was unresponsive to USP21-mediated deubiquitination (Supplementary Fig. 5d). These findings demonstrated that USP21 could stabilize FOXP3 through deubiquitination and prevented the depletion of FOXP3 protein in Treg cells.

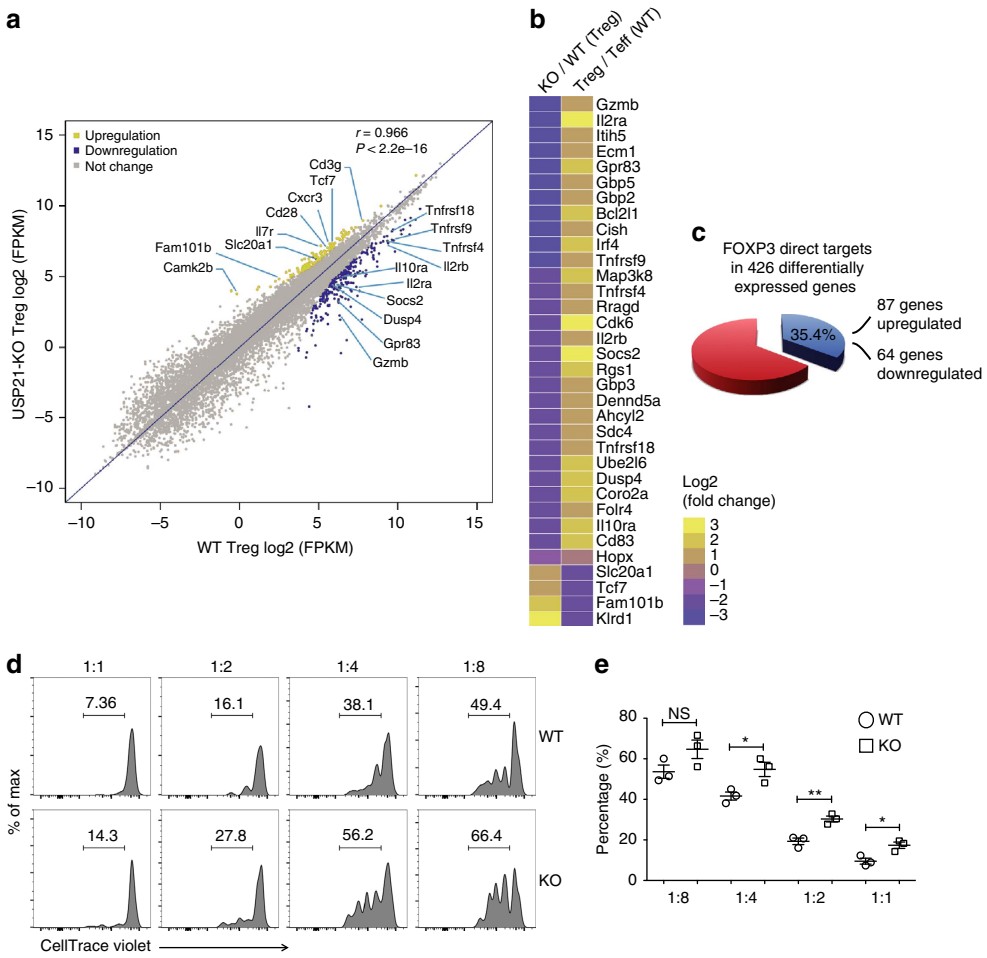

**Figure 5 | USP21-ΔTreg cells have compromised expression of Treg signature genes and impaired suppressive activity.** (**a**) Scatter plot of genes significantly upregulated (yellow), downregulated (blue) or unchanged in USP21-ΔTreg cells. The cutoff value was set at fold change ≥ 2 and $P$ value < 2.2e − 16. (**b**) Clustering of differentially expressed genes identified in USP21-ΔTreg cells compared with WT Treg cells. Yellow and purple represent high and low levels of expression of the indicated genes, respectively. The colours indicate the value of $\log_2$ fold change. (**c**) Percentage of FOXP3-bound genes that changed in expression in USP21-ΔTreg cells, where published FOXP3 ChIP-seq data sets were used as total FOXP3 target gene coverage. (**d**) CD4+CD25−YFP− Teff cells were sorted from WT mice and labelled with CellTrace Violet. Labelled responder T cells were then either cultured alone or mixed with varying amounts (as indicated) of CD4+CD25hiYFP+ Treg cells sorted from WT or KO mice. Cell mixtures were stimulated with anti-CD3/CD28 beads and the proliferation of the responder cells was measured three days later. (**e**) Percentage of proliferated responder T cells was assessed as shown in **d**. ($n = 3$ for each group). All data represent means ± s.d. *$P ≤ 0.05$, **$P ≤ 0.01$, as determined by Student's $t$-test. NS, not significant.

**USP21 is dispensable for GATA3 expression in murine Tregs.** USP21 was previously reported to positively regulate GATA3 expression through deubiquitination in human Treg cells[24]. In agreement with this previous study, we confirmed the endogenous interaction between USP21 and GATA3 in human expanding Treg cells (Supplementary Fig. 6a). However, GATA3 protein remained stable in murine USP21-ΔTreg cells from the thymus, spleen and peripheral lymph nodes (Supplementary Fig. 6b). We also compared the expression level of GATA3 protein in sorted CD4+CD25hiYFP+ Treg cells or polarized iTreg cells from WT and Usp21fl/flFoxp3Cre mice, and no significant difference was observed (Supplementary Fig. 6c). These results are consistent with a previous study that showed how USP21 is dispensable for the regulation of GATA3 during murine lymphocyte differentiation[23], perhaps suggesting the existence of a redundant role of USP21 in stabilizing GATA3 in mice, but not in humans.

**Discussion**

Ample evidence has supported the notion of how FOXP3 instability facilitates the generation of T-effector-cell-like Treg

cells[5,10,26–29]. Through generation of Usp21fl/flFoxp3Cre mice, we observed a population of Th1-like instable USP21-ΔTreg cells, which was accompanied by increased Th1 responses. This was due to increase in degradation of FOXP3 through the lack of deubiquitinase activity. These observations are consistent, albeit from an opposite mechanistic standpoint, with a previous finding that Stub1-overexpressing Treg cells became Th1-cell-like phenotype due to increased FOXP3 depletion via ubiquitination and proteasome-mediated degradation[19]. Moreover, in human patients with relapsing remitting multiple sclerosis (RRMS), a high frequency of IFN-γ-secreting Treg cells has been observed[26], confirming the existence of Th1-like Treg cells in humans. There have also been reports of how Treg cells with decreased FOXP3 expression preferentially become Th2-type effectors in hosts with severe Th2-type disorder[12]. In addition, under arthritic conditions, CD25loFOXP3+CD4+ T cells may lose FOXP3 expression and undergo transdifferentiation into Th17 cells[13]. Therefore, instable Treg cells may adopt various T-helper-cell-like phenotypes under different inflammatory contexts.

K48-linked polyubiquitinated FOXP3 is degraded in the proteasome through the action of the E3 ubiquitin ligase Stub1

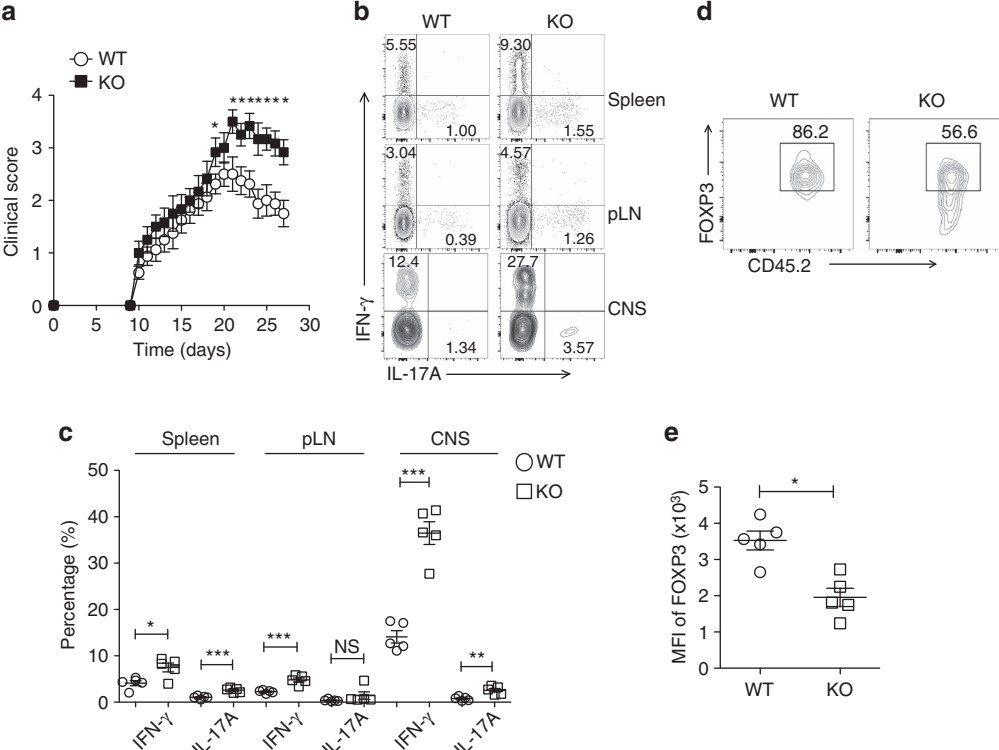

**Figure 6 | *Usp21*<sup>fl/fl</sup>*Foxp3*<sup>Cre</sup> mice developed more severe EAE symptoms.** (**a**) Clinical severity of EAE in WT ($n = 8$) and KO ($n = 7$) mice was monitored for 27 days after immunization with MOG peptide. Statistical analysis of the clinical signs was further described in Table 1. (**b**) Representative figure shown the expression of IFN-γ and IL-17A by CD4$^+$YFP$^-$ effector T cells in the spleen, peripheral lymph nodes and CNS from WT ($n = 5$) and KO ($n = 5$) mice on day 27 post EAE induction. (**c**) Percentage of IFN-γ$^+$ and IL-17A$^+$ cells among CD4$^+$YFP$^-$ effector T cells in the spleen, peripheral lymph nodes and CNS from WT ($n = 5$) and KO ($n = 5$) mice as indicated in **b**. (**d**) CD45.2$^+$ WT Treg or USP21-ΔTreg cells were transferred intravenously into CD45.1 mice at the onset of EAE (day 12). At the peak of EAE (day 17), the transferred CD45.2$^+$ Treg cells were analysed for FOXP3 expression in the CNS from the CD45.1 mice. (**e**) Mean fluorescent intensity (MFI) of FOXP3 among transferred CD45.2$^+$ Treg cells as indicated in **e**. ($n = 5$ for each group). All data represent means ± s.d. *$P \leq 0.05$, **$P \leq 0.01$, ***$P \leq 0.001$, as determined by Student's *t*-test. NS, not significant.

**Table 1 | Features of MOG$_{35-55}$-induced EAE.**

| Mice | Incidence (%) | Onset (day p.i.) | Peak (day p.i.) | Mean max. score |
|------|---------------|------------------|-----------------|-----------------|
| WT | 87 | 10 | 20 | 3.1 |
| KO | 83 | 10 | 21 | 3.75 |

KO, knockout; max., maximum; p.i., post immunization; WT, wild type.
Statistical analysis of the clinical signs in WT and KO mice, assessed as described in Fig. 6a.

(ref. 19). And another report also described how Stub1 and Cbl-b sequentially induce FOXP3 for ubiquitination and degradation[30]. Here USP21 prevents FOXP3 degradation most likely through removing K48-linked polyubiquitin moieties. Previously, another deubiquitinase, USP7, was also reported to stabilize FOXP3 through deubiquitination and improved Treg-cell functionality[20]. Therefore, these results collectively suggest that both deubiquitinases were required to maintain FOXP3 expression in Treg cells. In addition to FOXP3, USP21 might have additional targets in Treg cells, since a proportion of DEGs were not controlled directly by FOXP3. However, our data suggest that FOXP3 should be an important target of USP21 in Treg cells, since FOXP3 critically controls Treg-cell development and functional stability.

USP21-ΔTreg cells had compromised expression of Treg signature genes and displayed impaired suppressive activity. In an EAE disease model, we found that USP21-ΔTreg cells failed to restrict both Th1 and Th17 responses, possibly due to the loss of FOXP3 expression and subsequent loss of suppressive activity. Therefore, USP21-ΔTreg cells are largely hypoactive and failed to restrict the activation of inflammatory lymphocytes in lymphoid and non-lymphoid organs. Although CD4$^+$YFP$^-$ effector T cells produced IFN-γ at much higher levels in *Usp21*<sup>fl/fl</sup>*Foxp3*<sup>Cre</sup> mice, Th1-like USP21-ΔTreg cells could potentially contribute to disease pathology by amplifying Th1 responses through IFN-γ production and impaired effector-T-cell-mediated Th1 responses. To really understand whether this is the case, dual depletion of USP21 and effector-T-cell cytokines such as IFN-γ in Treg cells might be required.

Treg cells are known to restrict anti-tumour immune responses and promote tumour survival[31–35]. Therefore, temporary release from the suppressive influence of Treg cells or conversion of these cells into Th1-like inflammatory cells might provide beneficial traits in the tumour microenvironment to support anti-tumour responses. Interestingly, a recent study has characterized a population of FOXP3<sup>lo</sup> T cells in colorectal cancer, which produced higher amounts of IFN-γ after *in vitro* stimulation, also predicted better prognosis[36]. Therefore, Th1-like FOXP3<sup>lo</sup> Treg cells potentially contribute to anti-tumour immune responses and suppress tumour formation. Here the induction of FOXP3 degradation in Treg cells provides a potential method that enables transient inhibition of suppressive Treg cells, and thus USP21 is a potential drug target for future anti-cancer immunotherapy (Fig. 8).

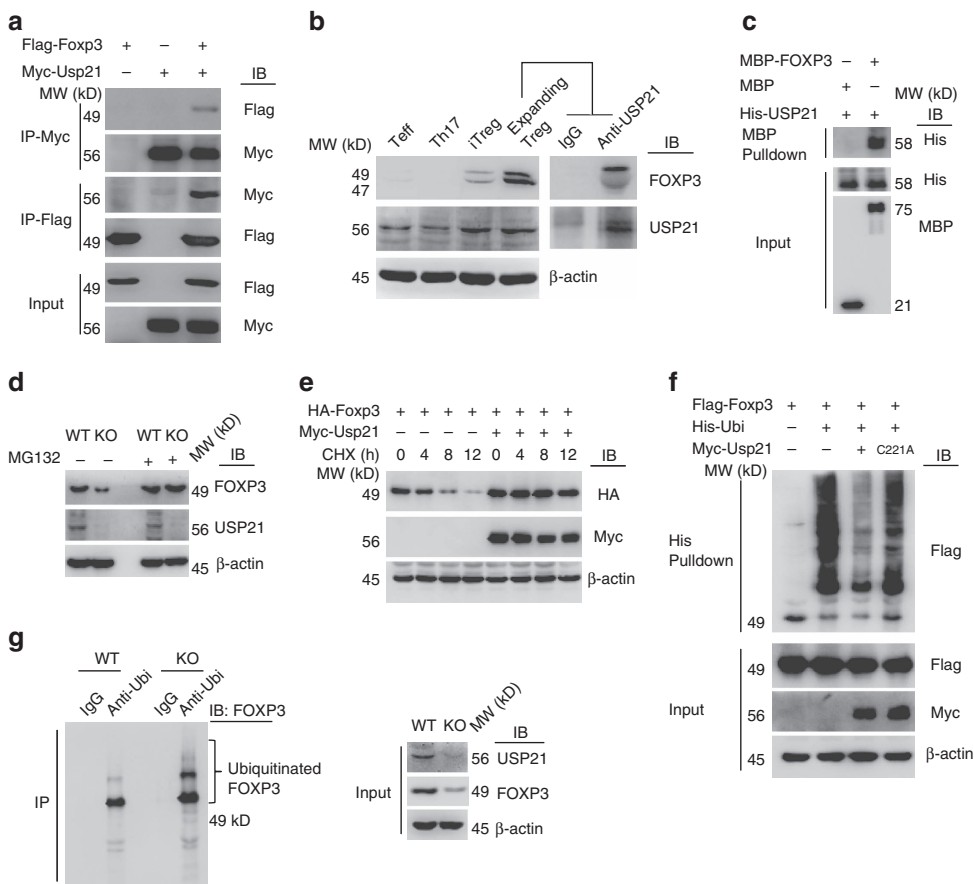

**Figure 7 | USP21 stabilizes FOXP3 through deubiquitination.** (**a**) Flag-FOXP3 and Myc-USP21 expression plasmids were cotransfected into 293T cells. Cells were lysed for co-IP as indicated. (**b**) Human CD4$^+$CD25$^{hi}$CD127$^{lo}$ expanding Treg cells were cultured with α-CD3 and α-CD28 antibodies in the presence of IL-2. Cells were collected for endogenous co-IP and western blotting. Cell lysates of Teff, Th17, iTreg and expanding Treg cells were loaded as input. (**c**) Purified His-USP21 and either MBP or MBP-FOXP3 were incubated together in a cell-free system. MBP pulldown assay was further performed followed by western blotting. (**d**) Polarized WT or KO iTreg cells were treated with or without MG132 for 4 h. Cell lysates were further analysed by western blotting. (**e**) HA-FOXP3 was expressed in HEK 293T cells with or without Myc-USP21. Cells were treated with 20 μg ml$^{-1}$ cycloheximide (CHX) for the indicated periods. Cell lysates were further analysed by Western blotting. (**f**) Pulldown assay of his-ubiquitin conjugated Flag-FOXP3 as described in methods. (**g**) Naive CD4$^+$ T cells were sorted from WT or KO mice and further polarized into iTreg cells. Cell lysates were immunoprecipitated with anti-Ubiquitin antibody and ubiquitinated FOXP3 levels were detected by western blotting.

## Methods

**Mice.** The mouse line carrying the *Usp21*$^{tm1a(EUCOMM)Wtsi}$ allele was purchased from the International Mouse Knockout Consortium. To generate *Usp21*$^{flox/flox}$ mice, *Usp21*$^{tm1a(EUCOMM)Wtsi}$ mice were further crossed with Flp recombinase transgenic mice of C57BL/6 strain. *Foxp3*$^{YFP}$-Cre mice were purchased from the Jackson Laboratory. Female *Foxp3*$^{Cre}$ (WT) and *Usp21*$^{fl/fl}$*Foxp3*$^{Cre}$ (KO) mice of C57BL/6 strain were used at 6- to 8-month old, unless otherwise noted. Randomization and blinding strategy was used whenever possible. All mice were maintained under specific pathogen-free conditions. All mouse experiments followed protocols approved by the Institutional Animal Care and Use Committee in Institut Pasteur of Shanghai.

**Antibodies.** Labelled anti-CD4 (GK1.5, 1:100), anti-CD8 (53-6.7, 1:100), anti-CD25 (PC61, 1:100), anti-CD44 (IM7.8.1, 1:100), anti-CD62L (MEL14, 1:100), anti-FOXP3 (FJK-16s, 1:100), anti-GATA3 (L50-823, 1:100), anti-IL-2 (JES6-5H4, 1:100), anti-IL-4 (BVD4-1D11, 1:100), anti-IL-17A (TC11-18H10, 1:100), anti-IFN-γ (XMG1.2, 1:100) and specific isotype-matched control antibodies (1:100) were from BD Biosciences, eBioscience or Miltenyl Biotec. Anti-USP21 (G-17, 1:1,000), Anti-Ubiquitin (P4D1, 1:1,000), Anti-GATA3 (HG3-31, 1:1,000) was from Santa Cruz Biotechnology. Anti-FOXP3 (eBio7979, 1:1,000) was from eBioscience.

**Flow cytometry.** For analysis of surface markers, cells were stained in PBS containing 2% fetal bovine serum (FBS) with antibodies as indicated. FOXP3 staining was performed according to the manufacturer's instructions (eBioscience). To determine cytokine expression, cells were stimulated with phorbol12-myristate 13-acetate (50 ng ml$^{-1}$), ionomycin (1 μM), Golgi Stop and Golgi Plug for 4 h.

At the end of stimulation, cells were stained with fixable viability dye eFluor 780 and the indicated antibodies according to the manufacturer's instructions (eBioscience).

**Cell purification and culture.** HEK 293T (ATCC CRL-11268) cells were cultured in DMEM containing 10% FBS and tested for mycoplasma contamination before use. Lymphocytes were isolated from lymphoid organs (spleen and peripheral lymph nodes) and CD4$^+$ T cells were enriched with isolation kits. CD4$^+$CD25$^{hi}$YFP$^+$ Treg cells and CD4$^+$CD25$^-$YFP$^-$ Teff cells were further sorted on a BD FACS ARIA II sorter (BD Biosciences). Sorted Treg and Teff cells were cultured in RPMI 1640 medium (plus β-mercaptoethanol) supplemented with 10% FBS (GIBCO), 1% GlutaMax (GIBCO), 1% sodium pyruvate (GIBCO), and 1% Pen/Strep (GIBCO). For *Usp21* knockdown assay, murine CD4$^+$CD25$^{hi}$YFP$^+$ Treg cells were transduced with virus containing shRNA lentiviral vector, along with anti-CD3/28 stimuli (1 cell to 1 bead). The following shRNA sequences were used: shCK, 5′-caacaagatgaagagcaccaa-3′; sh*Usp21*-1, 5′-ccagaaatacgtcccttcctt-3′; sh*Usp21*-2, 5′-cccagatgaaaggctcaagaa-3′. Mice CD4$^+$CD25$^{lo}$YFP$^-$CD62L$^+$ naïve T cells were sorted and further polarized into iTreg cells using anti-CD3/CD28 DynaBeads (Invitrogen) in the presence of mIL-2 (100 U ml$^{-1}$) plus mTGF-β1 (5 ng ml$^{-1}$, R&D). For Th1 conditions, 10 ng ml$^{-1}$ IL-12 and 10 μg ml$^{-1}$ anti-IL-4 were added. For Th17 conditions, 10 μg ml$^{-1}$ anti-IL-4, 10 μg ml$^{-1}$ anti-IFN-γ, 2 ng ml$^{-1}$ mTGF-β1, 20 ng ml$^{-1}$ IL-23 and 20 ng ml$^{-1}$ IL-6 were added. To enrich lymphocytes from the liver, lung and salivary glands, mice were transcardially perfused with heparin (10 U ml$^{-1}$) in PBS under anaesthesia. Tissues were excised independently of lymph nodes and were incubated for 1 h at 37 °C in Collagenase D (1 mg ml$^{-1}$; Roche), DNase (20 mg ml$^{-1}$, Roche) and 0.2% bovine serum albumin in RPMI 1640 medium. Digested tissue was strained through cell strainers. Lymphocytes were further separated by centrifugation through a 40–70% percoll gradient and then subjected to Flow Cytometry Analysis.

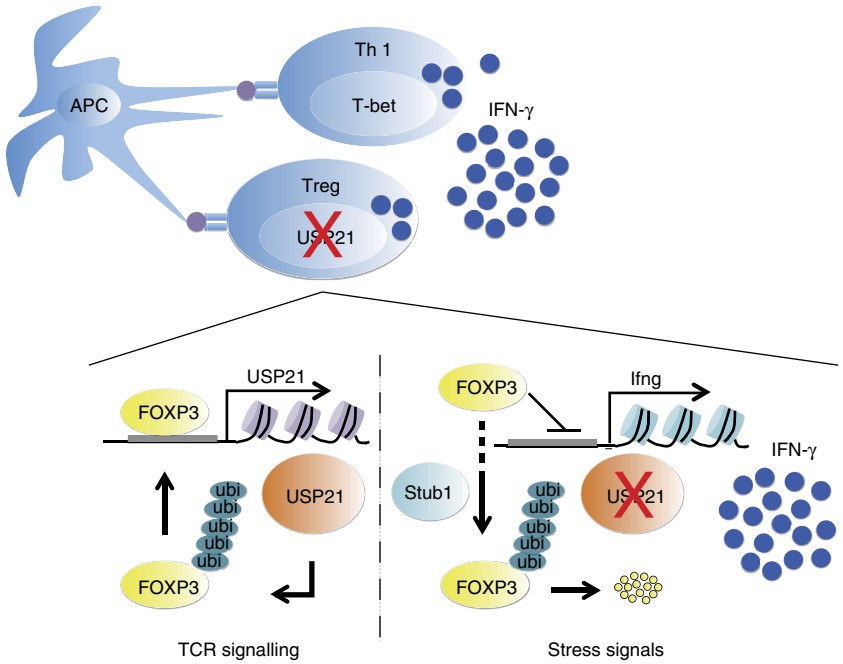

**Figure 8 | Working model describing the role of USP21 in preventing FOXP3 loss and Th1-like Treg-cell generation.** In Treg cells, FOXP3 activates the transcription of *Usp21* gene upon TCR stimulation. USP21 prevents the degradation of FOXP3 through deubiquitination and therefore controls the stability of Treg cells. Moreover, USP21-ΔTreg cells confer a Th1-like phenotype and facilitate Th1 responses in the host. Therefore, depletion of FOXP3hi Treg cells by using USP21 as the drug target potentially offers promise for anti-cancer immunotherapies.

Human CD4$^+$CD25$^{hi}$CD127$^{lo}$ Treg cells were isolated from PBMC by FACS on a BD FACS ARIA II sorter. And informed consent was obtained from healthy donors. All the studies were approved by the Institutional Ethics Committee. Human Treg cells were expanded using anti-CD3/CD28 DynaBeads (Invitrogen) at a cell to bead ratio of 1:3 in X-VIVO media (Lonza) supplemented with 10% FBS, 1% GlutaMax, 1% sodium pyruvate, and 1% Pen/Strep. Treg cells were expanded in the presence of rIL-2 (500 U ml$^{-1}$) and rapamycin (100 nM) for 10 days followed by resting in a lower concentration of rIL-2 (100 U ml$^{-1}$). Human CD4$^+$ CD25$^{lo}$CD45RA$^{hi}$ naive T cells were isolated from healthy donor PBMC and differentiated into iTreg cells using anti-CD3/CD28 DynaBeads (Invitrogen) in the presence of rhIL-2 (100 U ml$^{-1}$) and hTGF-β1 (5 ng ml$^{-1}$, R&D).

**EAE model.** Four-month old mice were immunized subcutaneously with 200 µl of emulsified CFA and 200 mg MOG$_{35-55}$ peptide (MEVGWYRSPFSRVVHLYRNGK), and received intraperitoneal injections of 200 ng pertussis toxin at the time of immunization, and repeated 48 h later. Animals were evaluated daily for signs of disease by the following criteria: 0, no disease; 1, flaccid tail paralysis; 2, hind limb paresis; 3, bilateral hind limb paralysis; 4, fore and hind limb paralysis.

**Adoptive transfer of Treg cells.** CD4$^+$CD25$^+$YFP$^+$ Treg cells were sorted from WT and KO mice. A total of 10$^6$ Treg cells were transferred intravenously into Rag1$^{-/-}$ mice. Rag1$^{-/-}$ mice were sacrificed 7 days later. A total of 2 × 10$^6$ CD45.2$^+$ WT Treg cells or USP21-ΔTreg cells were transferred intravenously into CD45.1 mice at the onset of EAE (day 12). At the peak of EAE (day 17), the transferred CD45.2$^+$ Treg cells were analysed for FOXP3 expression in the CNS from the CD45.1 mice.

**RNA preparation and immunoblotting.** Total RNA was extracted using TRIzol reagent (Invitrogen). cDNA was synthesized using a reverse transcriptase kit (TaKaRa, Japan), followed by qRT–PCR analysis (SYBR Green; TaKaRa). Cells were washed with ice-cold PBS and lysed on ice for 30 min in RIPA buffer (50 mM Tris-HCl, pH 7.5; 135 mM NaCl; 1% NP-40; 0.5% sodium DOC; 1 mM EDTA; 10% glycerol) containing protease inhibitor (1:100, P8340; Sigma-Aldrich), 1 mM NaF, and 1 mM PMSF. Cell lysates were cleared by centrifugation, and supernatants were immunoprecipitated with the appropriate antibodies (Abs, 1 µg ml$^{-1}$) using protein A/G-agarose beads at 4 °C. Samples were then used for immunoblotting analysis with indicated antibodies.

**His-ubiquitin pulldown assay.** Cells were lysed in a pH 8 urea buffer (8 M urea, 100 mM Na$_2$HPO$_4$, 10 mM TRIS (pH 8.0), 0.2% TX-100, 10 mM imidazole and 1 mM N-ethylmaleimide) and incubated with Ni-NTA beads for 2 h at room temperature. The beads were washed twice in pH 8 urea buffer, twice in pH 6.3

urea buffer (8 M urea, 100 mM Na$_2$HPO$_4$, 10 mM TRIS (pH 6.3), 0.2% TX-100 and 10 mM imidazole), and once in a wash buffer (20 mM TRIS (pH 8.0), 100 mM NaCl, 20% glycerol, 1 mM dithiothreitol and 10 mM imidazole). Samples were then used for immunoblotting analysis with indicated antibodies.

**MBP pulldown assay.** MBP, MBP-FOXP3 and His-USP21 were expressed in *Escherichia coli* and purified by amylose resin or Ni-NTA beads. His-USP21 were incubated with MBP or MBP-FOXP3 protein for 2 h at 4 °C in buffer A (20 mM Tris·HCl (pH 7.5), 200 mM NaCl, 5 mM β-mercaptoethanol, and 1 mM PMSF) and then were incubated with amylose resins for another 2 h. Amylose resins then were washed extensively with buffer A, followed by SDS/PAGE.

**RNA-Seq library generation.** Total RNA was isolated using RNeasy mini kit (Qiagen, Germany). Strand-specific libraries were prepared using the TruSeq Stranded Total RNA Sample Preparation kit (Illumina, USA) following the manufacturer's instructions. Briefly, mRNA was purified, fragmented and used for first- and second-strand cDNA synthesis followed by adenylation of 3′ ends. Samples were ligated to unique adaptors and subjected to PCR amplification. Libraries were quantified by Qubit 2.0 Fluorometer (Life Technologies, USA) and validated by Agilent 2100 bioanalyzer (Agilent Technologies, USA) to confirm the insert size and calculate the mole concentration. Cluster was generated by cBot with the library diluted to 10 pM and then were sequenced on the Illumina HiSeq 2500 (Illumina, USA). The library construction and sequencing was performed at Shanghai Biotechnology Corporation.

**High-throughput sequencing and analysis.** Sequencing raw reads were preprocessed by filtering out rRNA reads, sequencing adaptors, short-fragment reads and other low-quality reads. Tophat v2.0.9 was used to map the cleaned reads to the mouse mm10 reference genome with two mismatches. After genome mapping, Cufflinks v2.1.1 was run with a reference annotation to generate FPKM values for known gene models. Differentially expressed genes were identified using Cuffdiff. The *P* value significance threshold in multiple tests was set by the false discovery rate (FDR). The fold changes were also estimated according to the FPKM in each sample. The differentially expressed genes were selected using the following filter criteria: FDR ≤ 0.05 and fold change ≥ 2.

**Statistical analysis.** *P* values were calculated with Student's *t*-test or analysis of variance (GraphPad Prism) as specified in figure legends. All data represent means ± s.d. *$P \le 0.05$, **$P \le 0.01$, ***$P \le 0.001$, as determined by two-tailed, unpaired Student's *t*-test. NS, not significant. Sample sizes were designed with adequate power according to the literature and our previous studies. Randomization and blinding strategy was used whenever possible.

**Data availability.** The RNA-seq data that support the findings of this study are available in GEO database with the GEO accession number GSE77458. The uncropped scans of all western blots were supplied in Supplementary Figs 7 and 8. The authors declare that all the data supporting the findings of this study are available within the article and its Supplementary Information Files or from the corresponding author on reasonable request.

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

## Acknowledgements

Our research is supported by National Basic Research Program of China Grants 2014CB541803 and 2014CB541903; NSFC 30972702, 31170825, 31200647, 31200646, 81271835, 81270083, 81671611, 31670911, 31370863, 81330072; SMCST 14JC1406100; Shanghai Academic Research Leader 16XD1403800; Strategic Priority Research Program of the Chinese Academy of Sciences, Grant No. XDB19000000, XDB19040104; NIH-NSFC collaborative grant 81161120417; National Science and Technology Major Project 2012ZX10002007-003, 2013ZX10003009-002; NN-CAS Foundation. B.L. is a recipient of Shanghai 'Rising Star' program 10QA1407900, CAS '100-talent' program and National Science Foundation for Distinguished Young Scholars 31525008. A.T. is a recipient of NSFC grant 31150110337. We thank R. Zhu, P. Wang, Y-J. Dang, F. Pan and members of Li laboratory for helpful discussion.

## Author contributions

Y.L. designed and performed experiments, and wrote the manuscript; Y.L. contributed to cellular experiments; S.W. and F.Z. contributed to the technical support; A.T. contributed to scientific discussion, writing and editing the manuscript; B.L. designed experiments, contributed to writing the manuscript and provided overall direction.

## Additional information

**Competing financial interests:** The authors declare no competing financial interests.

