## [Peer Review File · Nature Communications]

Reviewers' comments:

Reviewer #1, expert in Treg biology (Remarks to the Author):

In this work Li and coworkers investigate the role of USP21 in the regulation of FoxP3+ Treg stability and function. With this goal in mind they generated conditional mice with a specific deletion of USP21 in Foxp3+ Tregs. They found signs of spontaneous autoimmunity, and worsened EAE in mice with USP21 deficiency in Tregs. The authors demonstrate decreased suppressive activity *in vitro*, correlated with decreased FoxP3 stability and expression of FoxP3-dependent genes. Based on these findings, they suggest that USP21 deficiency destabilize FoxP3+ Tregs. These are important studies, properly conducted. However it is my opinion that the authors should address some points before this work is ready for publication.

1. Transfer *in vivo* studies of WT and USP21 deficient foxp3 Tregs into naïve and EAE mice should be performed to demonstrate that the Foxp3+ Tregs are unstable *in vivo*, resulting in decreased suppressive activity.
2. Does USP21 affect the stability of other targets that may also be central for FoxP3+ Treg function (i.e. CD25). This is important, because USP21 may be action on multiple targets to regulate Tregs and not only on Foxp3.
3. Please show a table with proper statistical analysis for the EAE experiment shown in Fig 4f.
4. The authors should refer and discuss the Th1 like FoxP3+ Tregs described by Dominguez-Villar et al (Nat Med. 2011).

Reviewer #2, expert in ubiquitination in lymphocytes (Remarks to the Author):

This is a solid report by Li et al showing a Treg-intrinsic requirement for Usp21 for normal Foxp3+ Treg function. The authors show that Foxp3-specific cko mice develop spontaneous exaggerated IFN γ /Th1 type responses. Given this, it would be interested to the reader to know whether induction of peripheral Treg from conventional precursors is affected. Biochemical studies show convincingly that Usp21 deubiquitinylates Foxp3. However, studies of Foxp3 protein half-life in Treg or even Foxp3-transfected cells using chx block are warranted. In figure 3a the authors should gate on YFP-positive cells for their Foxp3 analysis, using an anti-YFP Ab if YFP fluorescence is extinguished by the fixation technique

Reviewer #3, expert in Treg biology (Remarks to the Author):

General

The manuscript "USP21 Prevents the Generation of T-helper-1-like Treg Cells" by Yangyang Li and colleagues claimed that E3 deubiquitinase USP21 stabilizes Foxp3 protein in Treg cells and prevent the generation of Th1-like Treg cells. Accumulating evidence indicate that Foxp3 is regulated by various post-transcriptional modifications, including ubiquitination, acetylation, and phosphorylation. The authors clearly indicated that USP21 directly interacts with Foxp3 and reduces its ubiquitination. The interesting point of this manuscript is to show that USP21 negatively modulates the skewing of Treg cells into Th1-like Treg cells. The authors' conclusion is interesting, but in some cases they are too speculative. Although they employed Foxp3 specific USP21 conditional knockout mice, one serious limitation of this study is that Foxp3 positive Treg are confused with activated effector T cells because Foxp3 is transiently up-regulated in all activated T cells. It is also still unclear how the ubiquitin-proteasome pathway is involved in this process. The significance of this study needs to be made more apparent. Jorg van Loosdregt et al have reported that Foxp3 protein expression is regulated by the

deubiquitinase USP7 (Immunity. 2013; 39(2): 259-271.). They also clearly indicated the polyubiquitination-mediated proteasomal degradation of Foxp3 protein is controlled by USP7. Thus, acceptance would be conditional on provision of additional data to support the current conclusions. Specific points in support of this recommendation are made below:

Major points

1. Line 154, as described above, the authors could not distinguish the effect of USP21 on Foxp3 positive Treg and activated Foxp3 positive effector T cells in USP21^{fl/fl}Foxp3^{YFP-Cre} mice. It is well known that activated T cells transiently express Foxp3 protein. Knockdown assay of UPS21 using siRNA or shRNA in WT Foxp3+ Treg cells may provide important information.

2. Line 101, it is not clear that the Th1 skewing in USP21^{fl/fl}Foxp3^{YFP-Cre} mice is due to cell intrinsic effect of USP21 deficiency in effector T cells or loss of suppressive function of USP21 deficient Foxp3+ Treg cells. To elucidate this point, authors should examine the in vitro differentiation of CD4+ T cell subsets (Th1, Th17, and Treg) using USP21^{fl/fl}Foxp3^{YFP-Cre} mice.

3. Line 188, although the authors demonstrated that ubiquitination of Foxp3 is regulated by USP21, it is not still unclear how the ubiquitin-proteasome pathway is involved in this process. To clarify the proteasomal degradation of ubiquitinated Foxp3, the authors should consider the use of proteasome inhibitor MG132 in Fig. 5. The stability of Foxp3 protein also should be evaluated by using a protein synthesis inhibitor cycloheximide.

Minor points

1. Line 88, please unify the name of USP21^{fl/fl}Foxp3^{YFP-Cre} mice (USP21^{fl/fl}Foxp3^{YFP-Cre}, USP21-dTreg, or USP21 KO).

2. Line 97 (Fig. 1b), the authors should show the data of spleen weights, because this is the only characteristic morphologic feature of USP21^{fl/fl}Foxp3^{YFP-Cre} mice.

3. Line 146, the authors should specify the number of upregulated and downregulated DEGs in Foxp3 Tregs from USP21^{fl/fl}Foxp3^{YFP-Cre} mice among 426 Foxp3 target genes. This is very important information to assess these RNA-seq data. Representative genes must be validated by qRT-PCR analysis.

4. The authors previously reported that Stub1-mediated Foxp3 ubiquitination occurred via Lysine-48 linkage (Immunity 2013;39(2)). The information of types of polyubiquitin linkage should be discussed more fully.

5. In the discussion, the authors should refer two papers described below.

E3 Ubiquitin Ligases Cbl-b Regulates Thymic-Derived CD4+CD25+ Regulatory T Cell Development by Targeting Foxp3 for Ubiquitination

Yixia Zhao et. al.

J Immunol. 2015; 194(4): 1639-1645.

Stabilization of the Transcription Factor Foxp3 by the Deubiquitinase USP7 Increases Treg-Cell-Suppressive Capacity

Jorg van Loosdregt et. al.

Immunity. 2013; 39(2): 259-271.

6. In the Materials and Methods, there is no information of RNA-seq. The authors should clarify the detailed methods such as library preparation, mapping, and DEG analysis.

Reviewer #1, expert in Treg biology (Remarks to the Author):

In this work Li and coworkers investigate the role of USP21 in the regulation of FoxP3+ Treg stability and function. With this goal in mind they generated conditional mice with a specific deletion of USP21 in Foxp3+ Tregs. They found signs of spontaneous autoimmunity, and worsened EAE in mice with USP21 deficiency in Tregs. The authors demonstrate decreased suppressive activity in vitro, correlated with decreased FoxP3 stability and expression of FoxP3-dependent genes. Based on these findings, they suggest that USP21 deficiency destabilize FoxP3+ Tregs. These are important studies, properly conducted. However it is my opinion that the authors should address some points before this work is ready for publication.

1. Transfer in vivo studies of WT and USP21 deficient foxp3 Tregs into naïve and EAE mice should be performed to demonstrate that the Foxp3+ Tregs are instable in vivo, resulting in decreased suppressive activity.

We appreciate and fully agree with our reviewer's insightful comments that it is essential to test the *in vivo* stability of USP21-deficient Treg cells. Therefore, we did additional experiments by transferring WT or USP21-deficient Treg cells into Rag1^{-/-} mice. We observed a significant loss of FOXP3 in *in vivo* transferred USP21-deficient Treg cells (Supplementary Fig. 4c-e). We next transferred CD45.2⁺ WT Treg or USP21-deficient Treg cells into EAE-bearing CD45.1 mice, and CD45.2⁺ USP21-deficient Treg cells became instable through FOXP3 loss (Supplementary Fig. 5e, f).

2. Does USP21 affect the stability of other targets that may also be central for FoxP3+ Treg function (i.e. CD25). This is important, because USP21 may be action on multiple targets to regulate Tregs and not only on Foxp3.

We fully agree with the reviewer's great comments. We have further discussed this important issue in the main text. In addition to FOXP3, USP21 might have additional targets in Treg cells, since a proportion of differentially expressed genes (DEGs) were not controlled directly by FOXP3. However, our data suggest that FOXP3 should be an important target of USP21 in Treg cells, since FOXP3 critically controls Treg cell development and functional stability. And we did observe an impaired transcription of *Ii2ra* (CD25) in USP21-deficient Treg cells (Supplementary Fig. 5a), possibly due to the loss of FOXP3.

3. Please show a table with proper statistical analysis for the EAE experiment shown in Fig 4f.

Thanks for this constructive suggestion. As suggested, we have added a table with proper statistical analysis for the EAE experiment in Fig. 4f.

4. The authors should refer and discuss the Th1 like FoxP3+ Tregs described by Dominguez-Villar et al (Nat Med. 2011).

We have added the following reference in the discussion part as suggested by our reviewer:

26 Dominguez-Villar, M., Baecher-Allan, C. M. & Hafler, D. A. Identification of T helper type 1-like, Foxp3⁺ regulatory T cells in human autoimmune disease. *Nature medicine* 17, 673-675, doi:10.1038/nm.2389 (2011).

Reviewer #2, expert in ubiquitination in lymphocytes (Remarks to the Author):

This is a solid report by Li et al showing a Treg-intrinsic requirement for Usp21 for normal Foxp3+ Treg function. The authors show that Foxp3-specific cko mice develop spontaneous exaggerated IFNg/Th1 type responses. Given this, it would be interested to the reader to know whether induction of peripheral Treg from conventional precursors is affected.

1. Biochemical studies show convincingly that Usp21 deubiquitinylates Foxp3. However, studies of Foxp3 protein half-life in Treg or even Foxp3-transfected cells using chx block are warranted.

We greatly appreciate this constructive comment from the reviewer. We next studied the half-life of FOXP3 using the protein synthesis inhibitor cycloheximide. And ectopically expressed USP21 significantly extended the half-life of FOXP3 protein (Supplementary Fig. 6d).

2. In figure 3a the authors should gate on YFP-positive cells for their Foxp3 analysis, using an anti-YFP Ab if YFP fluorescence is extinguished by the fixation technique.

We would like to thank our reviewer for this great question. Therefore, we checked the expression of FOXP3 in CD4⁺YFP⁺ Treg cells and confirmed the instability of FOXP3 protein in CD4⁺YFP⁺ USP21-deficient Treg cells (Supplementary Fig. 4a, b).

Reviewer #3, expert in Treg biology (Remarks to the Author):

The manuscript "USP21 Prevents the Generation of T-helper-1-like Treg Cells" by Yangyang Li and colleagues claimed that E3 deubiquitinase USP21 stabilizes Foxp3 protein in Treg cells and prevent the generation of Th1-like Treg cells. Accumulating evidence indicate that Foxp3 is regulated by various post-transcriptional modifications, including ubiquitination, acetylation, and phosphorylation. The authors clearly indicated that USP21 directly interacts with Foxp3 and reduces its ubiquitination. The interesting point of this manuscript is to show that USP21 negatively modulates the skewing of Treg cells into Th1-like Treg cells. The authors' conclusion is interesting, but in some cases they are too speculative. Although they employed Foxp3 specific USP21 conditional knockout mice, one serious limitation of this study is that Foxp3 positive Treg are confused with activated effector T cells because Foxp3 is transiently up-regulated in all activated T cells. It is also still unclear how the ubiquitin-proteasome pathway is involved in this process. The significance of this study needs to be made more apparent. Jorg van Loosdregt et al have reported that Foxp3 protein expression is regulated by the deubiquitinase USP7 (Immunity. 2013; 39(2): 259-271.). They also clearly indicated the polyubiquitination-mediated proteosomal degradation of Foxp3 protein is controlled by USP7. Thus, acceptance would be conditional on provision of additional data to support the current conclusions. Specific points in support of this recommendation are made below:

Major points

1. Line 154, as described above, the authors could not distinguish the effect of USP21 on Foxp3 positive Treg and activated Foxp3 positive effector T cells in USP21^{fl/fl}Foxp3^{YFP-Cre} mice. It is well known that activated T cells transiently express Foxp3 protein. Knockdown assay of UPS21 using siRNA or shRNA in WT Foxp3⁺ Treg cells may provide important information.

Thanks for this constructive suggestion. We performed the knockdown assay of *Usp21* in WT Treg cells and tested the suppressive activity of *Usp21*-silenced Treg cells. Knockdown of *Usp21* in WT Treg cells did impair their suppressive activity (Supplementary Fig. 5b-d), confirming that USP21 is required for Treg cell function.

2. Line 101, it is not clear that the Th1 skewing in USP21^{fl/fl}Foxp3^{YFP}-Cre mice is due to cell intrinsic effect of USP21 deficiency in effector T cells or loss of suppressive function of USP21 deficient Foxp3⁺ Treg cells. To elucidate this point, authors should examine the in vitro differentiation of CD4⁺ T cell subsets (Th1, Th17, and Treg) using USP21^{fl/fl}Foxp3^{YFP}-Cre mice.

We would like to thank our reviewer for this great question. Naïve CD4⁺ T cells were sorted from WT and *Usp21^{fl/fl}Foxp3^{Cre}* mice and then differentiated into Th1, Th17 and iTreg cells under polarizing conditions. We observed an increased production of IFN- γ in iTreg cells from *Usp21^{fl/fl}Foxp3^{Cre}* mice (Supplementary Fig. 2a), confirming that USP21- Δ Treg cells displayed a Th1-like phenotype. However, the production of both IFN- γ and IL-17A was comparable in polarized Th1 and Th17 cells from WT and *Usp21^{fl/fl}Foxp3^{Cre}* mice (Supplementary Fig. 2a). Therefore, Th1 skewing in *USP21^{fl/fl}Foxp3^{Cre}* mice is potentially due to aberrant function of Th1-like USP21- Δ Treg cells.

3. Line 188, although the authors demonstrated that ubiquitination of Foxp3 is regulated by USP21, it is not still unclear how the ubiquitin-proteasome pathway is involved in this process. To clarify the proteasomal degradation of ubiquitinated Foxp3, the authors should consider the use of proteasome inhibitor MG132 in Fig. 5. The stability of Foxp3 protein also should be evaluated by using a protein synthesis inhibitor cycloheximide.

We greatly appreciate this constructive comment from our reviewer. To clarify the proteasomal degradation of ubiquitinated FOXP3, we treated USP21-deficient Treg cells with the proteasome inhibitor MG132. We found the loss of FOXP3 protein in USP21-ΔTreg cells could be prevented by the addition of proteasome inhibitor MG132 (Supplementary Fig. 6c), suggesting that the ubiquitin-proteasome pathway is involved. We also studied the half-life of FOXP3 using the protein synthesis inhibitor cycloheximide. And ectopically expressed USP21 significantly extended the half-life of FOXP3 protein (Supplementary Fig. 6d).

Minor points

1. Line 88, please unify the name of USP21^{fl/fl}Foxp3^{YFP-Cre} mice (USP21^{fl/fl}Foxp3^{YFP-Cre}, USP21-dTreg, or USP21 KO).

As suggested, we have unified the name of Usp21^{fl/fl}Foxp3^{Cre} mice in our manuscript.

2. Line 97 (Fig. 1b), the authors should show the data of spleen weights, because this is the only characteristic morphologic feature of USP21^{fl/fl}Foxp3^{YFP-Cre} mice.

Thanks for this constructive suggestion. Please see the data of spleen weights in Fig. 1c.

3. Line 146, the authors should specify the number of upregulated and downregulated DEGs in Foxp3 Tregs from USP21^{fl/fl}Foxp3^{YFP-Cre} mice among 426 Foxp3 target genes. This is very important information to assess these RNA-seq data. Representative genes must be validated by qRT-PCR analysis.

Thanks for this constructive suggestion. As suggested, we have added the number of upregulated and downregulated FOXP3 targeting DEGs in Fig. 4c. There were 151 FOXP3 targeting DEGs in USP21- Δ Treg cells, including 87 upregulated DEGs and 64 downregulated DEGs (Fig. 4c). And the expression of representative genes was further validated by qRT-PCR (Supplementary Fig. 5a).

4. The authors previously reported that Stub1-mediated Foxp3 ubiquitination occurred via Lysine-48 linkage (Immunity 2013;39(2)). The information of types of polyubiquitin linkage should be discussed more fully.

As suggested, we have further discussed this important issue in the main text as following:

"K48-linked polyubiquitinated FOXP3 is degraded in the proteasome through the action of the E3 ubiquitin ligase Stub1. And another report also described how Stub1 and Cbl-b sequentially induce FOXP3 for ubiquitination and degradation. Here, USP21 prevents FOXP3 degradation most likely through removing K48-linked polyubiquitin moieties."

5. In the discussion, the authors should refer two papers described below.

E3 Ubiquitin Ligases Cbl-b Regulates Thymic-Derived CD4+CD25+ Regulatory T Cell Development by Targeting Foxp3 for Ubiquitination.

Yixia Zhao et. al J Immunol. 2015; 194(4): 1639-1645.

Stabilization of the Transcription Factor Foxp3 by the Deubiquitinase USP7 Increases Treg-Cell-Suppressive Capacity.

Jorg van Loosdregt et. al Immunity. 2013; 39(2): 259-271.

Thanks for our reviewer's kindly suggestion. We have added the following references in the discussion:

20 Van Loosdregt, J. et al. Stabilization of the transcription factor Foxp3 by the deubiquitinase USP7 increases Treg-cell-suppressive capacity.

Immunity 39, 259-271, doi:10.1016/j.immuni.2013.05.018 (2013).

- 30 Zhao, Y. et al. E3 Ubiquitin Ligase Cbl-b Regulates Thymic-Derived CD4+CD25+ Regulatory T Cell Development by Targeting Foxp3 for Ubiquitination. J Immunol 194, 1639-1645, doi:10.4049/jimmunol.1402434 (2015).

6. In the Materials and Methods, there is no information of RNA-seq. The authors should clarify the detailed methods such as library preparation, mapping, and DEG analysis.

Thanks for this great suggestion. In the methods section, we have added the detailed information of RNA-seq, including RNA-Seq library generation, high-throughput sequencing and analysis.

REVIEWERS' COMMENTS:

Reviewer #1 (Remarks to the Author):

The authors have addressed all my comments.

Reviewer #2 (Remarks to the Author):

The authors have addressed my concerns to an acceptable degree. The authors have appended all the new experimental data requested by the reviewers as supplemental figures, however, many of these findings are integral to the the conclusions of the paper, and should be integrated into the main figures.

Reviewer #3 (Remarks to the Author):

This second version of the manuscript "USP21 Prevents the Generation of 1 T-helper-1-like Treg Cells" by Yangyang Li and colleagues has been revised well. The improved manuscript contains sufficient information to merit its publication in the Nature Communications. I think this manuscript is now acceptable for publication.

Reviewer #2 (Remarks to the Author):

The authors have addressed my concerns to an acceptable degree. The authors have appended all the new experimental data requested by the reviewers as supplemental figures, however, many of these findings are integral to the conclusions of the paper, and should be integrated into the main figures.

Thank you for this constructive suggestion. As suggested, we have integrated some supplemental data into main figures. And these changes can be found in our new main figures (Fig. 4 to Fig. 8).